# Understanding responsibility for health inequalities in children's hospitals in England: a qualitative study with hospital staff

Liz Brewster ![ORCID],[1] Louise Brennan,[1] Avni Hindocha ![ORCID],[1,2] Judith Lunn ![ORCID],[1] Rachel Isba ![ORCID] [1]

[1]Lancaster Medical School, Lancaster University, Lancaster, UK
[2]Royal Manchester Children's Hospital, Manchester University NHS Foundation Trust, Manchester, UK

**Correspondence to**
Dr Liz Brewster;
e.brewster@lancaster.ac.uk

## ABSTRACT

**Objectives**  This study aimed to understand how staff in children's hospitals view their responsibility to reduce health inequalities for the children and young people who access their services.

**Design**  We conducted an exploratory qualitative study.

**Setting**  The study took place at nine children's hospitals in England.

**Participants**  217 members of staff contributed via interviews and focus groups conducted January–June 2023. Staff were represented at all levels of the organisations, and all staff who volunteered to contribute were included in the study.

**Analysis**  Data were analysed using Rapid Research Evaluation and Appraisal (RREAL) methodology for rapid assessment procedures (RAP).

**Results**  All of the children's hospitals were taking some action to reduce health inequalities. Two key themes were identified. First, it was clear that reducing health inequalities was seen as something that was of vital import and should be part of staff's day-to-day activity, framed as 'everyone's business.' Many staff felt that there was an obligation to intervene to ensure that children and young people receiving hospital treatment were not further disadvantaged by, for example, food poverty. Second, however, the deeply entrenched and intersectional nature of health inequalities sometimes meant that these inequalities were complex to tackle, with no clear impetus to specific actions, and could be framed as 'no-one's responsibility'. Within a complex health and social care system, there were many potential actors who could take responsibility for reducing health inequalities, and staff often questioned whether it was the role of a children's hospital to *lead* these initiatives.

**Conclusions**  Broadly speaking, senior leaders were clear about their organisational role in reducing health inequalities where they impacted on access and quality of care, but there was some uncertainty about the perceived boundaries of responsibility. This led to fragility in the sustainability of activity, and a lack of joined-up intervention. Most hospitals were forging ahead with activity, considering that it was more important to work to overcome health inequalities rather than debate whose job it was.

## STRENGTHS AND LIMITATIONS OF THIS STUDY

⇒ This is the first study to investigate staff's views on health inequalities in children's hospitals in England.

⇒ The sample size of over 200 participants created a substantial dataset for qualitative analysis, which indicates that the findings are robust and transferrable.

⇒ Participants were self-selecting and so may have volunteered to participate because they had a particular interest in sharing views on health inequalities.

## INTRODUCTION

Definitions of health inequalities vary, but there is broad agreement that these inequalities are unfair and avoidable.[1] These inequalities are also systemic or structural rather than purely individual.[2 3] Health inequalities are experienced on a social gradient, with those who are more socioeconomically disadvantaged facing greater health challenges, in experiences of illness and access to healthcare.[4] In the UK, Michael Marmot and the Institute of Health Equity have been key contributors to driving forward an agenda that recognises the impact of health inequalities on children and young people.[5 6] In 2010, the publication of the Marmot Review 'Fair Society, Healthy Lives'[7] set out a framework for action that emphasised how these inequalities accumulate across the lifespan. The recommendations prioritised children as a population group requiring early and meaningful intervention to prevent health inequalities becoming entrenched. This review was updated in 2020, presenting evidence that the gap in health status and outcomes between affluent and socioeconomically deprived areas had increased rather than decreased.[8]

Children living in areas of high deprivation are more likely to have worse health outcomes[9] and long-term conditions like

asthma.[10] Access to care also differs, with those from the most deprived areas being less likely to be brought to outpatient appointments.[11] Health inequalities therefore represent a tangible issue that needs to be addressed internationally, and have a direct impact on the effectiveness of health service provision.

In the UK, healthcare is publicly funded and provided free at the point of use, via the National Health Service (NHS). NHS organisations have a statutory duty with regards to specific equality domains defined by the Equality Act 2010,[12] which outlined protected characteristics (eg, age, sex, race) but there had not been a specific duty on NHS organisations to intervene with regard to health inequalities caused by socioeconomic deprivation until the Health and Social Care Act 2012,[13] which introduced the first legal duties relating to health inequalities. Health inequalities are also part of several subsequent national strategies, for example, the NHS Long Term Plan[14] and the NHS People Plan.[15] These top-level strategies focus on a legal duty to reduce inequalities in access, and outcomes for all without specific guidance on how this may be implemented.

Both socioeconomic and health inequalities are thought to have increased since the SARS-CoV-2/COVID-19 pandemic, impacting health outcomes, with longer waiting times for elective care. Greater attention has been given to reducing inequalities because of this increase, and the impact on children and young people has been identified as a particular focus for intervention, via initiatives such as NHS England's 'Core20PLUS5' approach.[16] This recognises that inequalities are intersectional, focusing on the most deprived 20% of the population, prioritising inclusion, for example, of people from ethnic minority communities, and identifying five clinical areas of focus: asthma, diabetes, epilepsy care for those with a learning disability, oral health and mental health.

Within the NHS, children's hospitals provide specialist paediatric care to children and young people, typically up to the age of 16. This provision can be as a stand-alone hospital for children, or as part of a wider organisational structure (NHS trust). There is great diversity among service provision (including among the sites in this study) because of these organisational structures, but in the main children's hospitals offer services (eg, emergency and acute general medicine, community and outpatient care) to their local population, and then specialist services to patients with more complex medical needs (eg, paediatric cardiac surgery) across a regional or national catchment. Children and young people and their caregivers may therefore travel a great distance to access these more specialist services.

This study aimed to understand how staff in children's hospitals in England view their responsibility to reduce health inequalities for the children and young people they serve.

## METHODS

Qualitative data were collected via interviews and focus groups, working across nine children's hospitals in England, as part of a wider research project (online supplemental file 1). The qualitative approach was selected to explore participants' understanding of current policy and practice around socioeconomic inequalities and their impact on healthcare, rather than hypothesising about, or predicting, responses.[17]

Rapid Research Evaluation and Appraisal (RREAL) methodologies[18] were used to collect and analyse data; these methodologies emerge from a qualitative research tradition that recognises the need for rapid data analysis to produce meaningful recommendations for practice in a timely manner. The full process had four stages, in which data were collected, recorded, synthesised and analysed to produce key recommendations. Using multiple population groups (senior leaders, doctors, nurses, allied health professionals, and professional/support staff), multiple data sources (interviews and focus groups) and multiple experienced researchers allowed us to ensure rigour by triangulating insights.[19]

### Recruitment and participants

Hospitals sampled are all members of the Children's Hospital Alliance, which commissioned a broader research project to exploring the roles that their member organisations can play in reducing socioeconomic barriers to paediatric care, and in understanding and developing best practice to do this. At time of the study, the Children's Hospital Alliance had 10 members; it now has 11. As a national network of specialist NHS trusts focused on paediatric care in the UK, it initially focused on support for pandemic recovery for children and young people. One of the original 10 member organisations was unable to participate in this study due to capacity. The hospitals varied in geographical location, percentage of local population living in the most 20% of deprived communities, organisational size and structure and services offered to the paediatric population.

Recruitment was organised by the nine participating organisations who circulated a call to volunteer to participate and organised access to rooms/teams on site. In total, 217 members of staff were included across the sites (table 1).

Staff were represented at all levels from senior executives to housekeeping staff. Focus groups were composed of staff from similar categories to ensure that those present felt more comfortable to contribute. To maintain anonymity when presenting data, we refer to broad three categories of staff:

► Leadership (n=47): executive and non-executive directors, anyone involved in the management of initiatives concerned with health inequalities.
► Clinical (n=85): doctors, nurses and allied health professionals (eg, physiotherapists, occupational therapists).
► Professional and support staff (n=85): for example, receptionists, porters, security and housekeeping.

**Table 1** Staff participants across nine NHS organisations

| Organisation | Number of participants in interviews | Number of focus groups/conversations and number of participants | Total number of participants per site |
| --- | --- | --- | --- |
| Site A | 9 | 4 groups, 19 participants | 28 |
| Site B | 4 | 4 groups, 29 participants | 33 |
| Site C | 7 | 2 groups, 8 participants | 15 |
| Site D | 10 | 4 groups, 12 participants | 22 |
| Site E | 4 | 2 groups, 15 participants | 19 |
| Site F | 11 | 15 'walk arounds'; 42 participants | 53 |
| Site G | 6 | 1 group, 5 participants | 11 |
| Site H | 5 | 2 groups, 15 participants | 20 |
| Site I | 5 | 2 groups, 11 participants | 16 |
| **Total number of participants across sites** | **61** | **39** groups/conversations | **217** |

NHS, National Health Service.

## Data collection

The research team consisted of two academics with expertise in applied qualitative health research and three research-active public health specialists. We conducted semistructured interviews with senior leaders and doctors with an interest in health inequalities across nine organisations. Interviews lasted between 30 and 60 min. A structured topic guide (online supplemental file 2) was used to provide an initial focus and allow for comparability between respondents, while also allowing for flexibility and individuality in the responses given. Interviews were recorded via Microsoft Teams. In total, 61 interviews were completed, and number of interviews per organisation ranged from 4 to 11.

We conducted focus groups and had informal conversations with clinical, professional and support staff. The majority of focus groups were audio recorded, but for informal conversations, brief written notes with no identifiable details were made and then team members audio recorded a debrief session to provide an overview of the conversations. One or two focus groups were conducted per organisation, lasting around 60 min per group. Number of participants ranged from 3 to 10 members of staff. A structured topic guide (online supplemental file 3) provided focus for discussion. One site advised that informal 'walk around' conversations were more appropriate to engage staff; these discussions were also structured using the same topic guide. Written informed consent was taken for participation in interviews and focus groups, and verbal consent was taken for 'walk around' discussions. All nine organisations were visited for 1 or 2 days, meaning additional debrief recordings were often comprehensive and detailed. In total, 39 focus groups and informal discussions were completed. Data were collected between January and June 2023 with full ethical approvals. Participants gave informed consent to participate in the study before taking part.

## Patient and public involvement

As part of our wider ongoing programme of work, we have engaged with children and young people and their families to understand barriers to engagement with research around health inequalities.[20] This engagement work was conducted concurrently with the study reported here but did not inform data collection or analysis.

## Data analysis

Data analysis was conducted in several stages, informed by the RREAL methodology for rapid assessment procedures (RAP).[21] The RAP process is considered an appropriate method to conduct rigorous and robust qualitative data analysis in a short amount of time. It is predominantly team based and collaborative in nature, and conducted as an iterative process alongside data collection. Data are independently reviewed by multiple team members, before being brought together in a group session for further analysis. RAP is particularly suitable for this type of research, which aims to synthesise diverse perspectives on a defined topic, and is focused on generating useful learning that can be actioned. While it was felt that 'data saturation'[22] was being reached at organisations, it is evident that results are intrinsically linked with the sample of people that we spoke to at each hospital.

Following the first stage of data collection, three further stages (multiple RAP sheet generation, individual consolidation and group confirmation) completed the RREAL process. After each interview, focus group, or debrief, a structured RAP sheet (online supplemental file 4) outlining the main themes was completed by the researcher leading the data collection. One or more members of the research team then listened to the recording, and independently completed a RAP sheet. RAP sheet generation occurred simultaneously with data collection, with discussion occurring after data collection had been completed. A minimum of two standardised

RAP sheets was completed for each interview/focus group, and used to guide our reflexive analysis process. In total, 171 RAP sheets were generated. Researchers then read and reviewed all the RAP sheets generated, and completed independent informal notes to support synthesis. Two members of the team independently generated further 'site-specific' RAP sheets, providing an overview of key themes at each organisation. This synthesis process focused on considering the specific context of each organisation as well as the broader project research questions.

We then conducted several day-long group debriefing and analysis sessions, in which we focused on identifying key themes from RAP sheets. This process led to the creation of brief data summaries that were then used iteratively for cross-case comparison. The whole team contributed to these sessions, which aimed to establish consensus, while still being sensitive to diversity of views and experiences in different organisations. Each analysis session ended with confirmation and agreement of key themes.

## RESULTS

From our qualitative dataset, including the perspectives of 217 staff across 9 organisations, 2 main themes were identified that help to understand how staff in children's hospitals in England view their responsibility to reduce health inequalities for the children and young people using their services. The first was that staff saw reducing health inequalities as 'everyone's business' and that no child should experience inequalities in care or outcomes because of their socioeconomic status. This was particularly visible in terms of the activities that hospitals were undertaking in this space, regardless of national steer and guidance.

However, the second theme—while also recognising that health inequalities were deeply entrenched and intersectional—suggested that these inequalities were seen as 'no-one's responsibility' or 'not part of core business' for a children's hospital. Within a complex health and social care system, there were many potential actors who could take responsibility for reducing health inequalities, and staff often questioned whether it was the role of a children's hospital to lead these initiatives rather than treating the presenting illness. This led to some fragility in the sustainability of initiatives.

Nevertheless, all of the children's hospitals were taking some action to reduce health inequalities. Many staff felt that there was an obligation to intervene to ensure that children and young people receiving hospital treatment (and their caregivers) were not further disadvantaged by, for example, food poverty.

### Health inequalities as 'everyone's business'
The theme of preventing health inequalities as something that should be 'everyone's business' was observed in discussion of the strategy and day-to-day activities of each organisation. Presentation of this theme considers how 'responsibility' was understood in relation to organisational priorities, and the challenges to taking up this responsibility. We pay particular attention to communication, formal and informal support, and the impact of the pandemic on providing additional evidence of the impact of these inequalities, which built the case for intervention.

All participants were asked about their knowledge of organisational policies and strategies that related to health inequalities. Over half of the nine organisations already had a health inequalities strategy in place, although for larger organisations who also offered adult provision, this was an organisation-wide strategy rather than child focused. The remaining organisations all discussed making progress towards a strategy. The majority of organisations had a health inequalities steering group and/or forum. More than half had a data dashboard, or made reference to data analysis activity, and this was used to try to identify patients at risk of not attending appointments. This activity was led at an organisational level, rather than linked to national strategy.

Some participants were clear about the organisational strategy around health inequalities, but others were less certain what the overall vision was, or what activities connected to it. In some organisations, there was a named individual with responsibility for health inequalities, but this was not universal, and often, people working on the 'shop floor' were less familiar with the strategic direction. We found that awareness varied in terms of staff knowledge of strategy and policy around health inequalities. While this finding was heavily dependent on the sample asked, each site had some staff who were unaware of strategy or activity. It became clear that there was a lack of clear communication about work on health inequalities.

> It's frustrating as a clinician… I asked my leads and they said they didn't know, there's all these layers of people who didn't know… eventually I found out about a steering group. But it's not affecting my practice what they're doing [about reducing inequalities]. (Clinical, site B)

This lack of communication was one example of how work to reduce health inequalities was fragile. It was often led by keen individuals, and so while health inequalities were seen as something that should be a focus for activities ('everyone's business'), this could be more at the level of a vision that informed activities rather than something with concrete key performance indicators.

Consideration of reduction of health inequalities was clearly visible in interventions led by the organisations. There were several common forms of support across organisations, almost all of these focused on meeting day-to-day basic needs. These included cheap food offers, free food for breastfeeding mothers, free sanitary products, parking/travel reimbursement and hospital transport for specific identified groups, accommodation and interpreters. While available in the majority of organisations, these basic offers were not universal across all

sites and specialties. Other initiatives which occurred less frequently included food vouchers, a universal patient transport offer and free food on wards for household members.

As well as strategic direction and organisationally initiated formal support for families, a second, more hidden layer of support was present in many of the organisations. This informal system was largely driven by individuals, often at nursing or domestic support level, and showed how reducing health inequalities permeated into every level of work. Stories of housekeeping staff taking washing from families, buying caregivers coffees and food, and giving leftover food to families were frequent. Similarly, nursing staff discussed ways to order additional food when they noticed someone was in need. In some hospitals, ward managers had access to a budget that funded food vouchers and taxis. This support was partly dependent on motivated and caring individuals.

> I think the people that I speak to, and I speak to a broad range, it's actually part of their core value rather than something that they think would make a good headline. And I think that resonates throughout the entire organization that they would be doing this even if no one was watching them do it. (Leadership, site F)

Some staff felt that there needed to be key performance indicators around health inequalities to encourage prioritisation of resource to this area of activity. With no current metrics around health inequalities, it becomes one of the 'nice to haves' rather than an essential.

> We don't have standards for the level of support for our families and we should do shouldn't we? (Leadership, site D)

> We can't allow health inequalities to become a luxury activity for organisations that happen to have the resources. (Leadership, site G)

Some staff explained that while health inequalities were not new, the pandemic had highlighted inequalities in outcomes and provided evidence (and therefore understanding) of the health inequalities that passionate members of staff had been trying to explain previously.

> I'm sure if you'd have talked about some of this stuff five years ago people wouldn't have understood as much as they do now. (Clinical, site A)

It was felt that the pandemic had also shone a focus on inequalities, kickstarting projects such as analysis of data on potential inequalities.

> There's variation in how long certain groups are waiting for care… we've observed in our analysis those with the lowest income wait on average six years longer. Children and young people with a learning disability wait longer to complete their treatment. Both of those factors were there before Covid, but now we

have done the analysis and are looking at how we can address it. (Leadership, site B)

Most of the leadership and clinical staff we interviewed were vocal about the way that health inequalities were their business and affected their day-to-day provision of healthcare.

> Because if they don't have a good start, they're not gonna have a long and happy life. So they will have poor diet. Doesn't matter what people tell you. It is quite expensive to eat well, so they will have increased diabetes. They will have increased obesity. They'll have all the pitfalls of living in a cold, damp house. So they'll have worse asthma. (Leadership, site F)

> Making people realise that being curious about why a certain child keeps coming back is just as much my responsibility as a doctor as it is to prescribe the Ventolin Inhaler. (Clinical, site G)

This idea of 'prevention rather than cure' was epitomised in several examples offered of activity to reduce health inequalities. For example, one clinician discussed how the impacts of having a domestic violence prevention service piloted in their Paediatric Emergency Department might have wide-ranging impacts for children and young people attending the department, but that those effects might be seen in educational attainment in a generation's time rather than being immediately visible.

### Health inequalities as 'no-one's responsibility'

In contrast to the above discussion of health inequalities as a priority area when inequalities impacted access and quality of care, this theme highlights how staff varied in their articulation of this priority in practice. Unclear boundaries between the hospital and other services, a lack of a national steer, challenges around wider system connections, concerns about potential cost and a lack of 'embeddedness' to make tackling inequalities part of core work were seen as challenges.

Although, broadly speaking, members of leadership teams were clear about their role in overcoming socioeconomic inequalities where they impacted on access and quality of care, there was some uncertainty about the perceived boundaries of responsibility with other institutions (eg, local government, other healthcare services). Some staff saw that leading the health inequalities agenda was not the role of the hospital, and there was an awareness that other services such as local authorities, public health, schools, community care and general practice also had a role to play.

> So whose responsibility is it to do this? One answer is everyone's responsibility. Another answer is not ours. And our job is here to see and treat people in front of us. That is one response. And you think well, there are public health teams around. There's lots of stuff. Should we be out there in the city centre running our campaigns about how to brush teeth? Or should we

put a dentist in theatre doing the dental extractions? That's the choice. (Leadership, site F)

Staff at all levels felt that a key barrier to addressing health inequalities action was the lack of national and local priority given to the subject. Senior executives were honest that priority was often given to the national targets such as waiting lists and finances.

It's not the Government's priority. No government is interested in long-term gain. They're interested in waiting lists and how they got better during their jurisdiction. (Leadership, site D)

Although some senior staff spoke about working as part of a system, staff members at all levels, and across all organisations, spoke about their frustrations in not feeling part of the system. This was seen as a barrier to advancing the inequalities agenda for children and young people.

It's being holistic as a system that we're not very good at doing. Because we are separate to social services and we're separate to schools, we don't think of 'children's' as a system yet. (Leadership, site D)

This was compounded by the inability of IT systems to link up the health and social care system.

No one has time or capacity to make these sort of links happen. It is just not at the top of their agenda to try and get everyone equitable access in one area. You can see a massive discrepancy across the region and actually it does have an impact on death rates in our service. (Clinical, site I)

For some organisations, their status within the community as a 'trusted brand' meant that they felt it was within their remit to intervene, regardless of their 'responsibility.' Children's hospitals were seen as well placed to lead on and communicate on health inequalities to the wider public.

We've got a trusted brand and … it's well known, it's well understood. And if we went walking up and down the high street with donation boxes, people go oh yeah, I'll give money to the children's hospital. (Leadership, Site F)

The picture was more mixed in terms of the views of the broader staff cohort. Some staff saw support for overcoming health inequalities as part of their role, but others felt it was not a priority, or felt less confident in asking questions about deprivation and need. Most health inequalities work was being driven by passionate individuals, making it unsustainable if these people leave. These champions were a positive, in terms of driving action, but their impact was limited by the way in which health inequalities were not part of core business or communicated within their organisations, and often went unnoticed by staff at leadership level. The lack of embeddedness meant that even if the aim was for every member of staff to take responsibility for tackling the

health inequalities they encountered, they were not held responsible for delivering this outcome.

A smaller number of staff were concerned about the overall expense of initiatives such as free food and transport, particularly if they were promoted widely. They did not see support for patients as a key role for the organisations, and highlighted the potential for initiatives to be inappropriately accessed by people who did not need the support. They also recognised the impact of increased workload, for example, if families were fed alongside patients, and noted that this could be time-consuming. It became clear that for inequalities to become part of core business, a significant shift was required.

## DISCUSSION

This paper has presented findings from a large-scale qualitative study aiming to understand how staff in children's hospitals in England view their responsibility to reduce health inequalities for the children and young people who access their services. Key findings were that while reducing health inequalities was broadly seen as important to these organisations, there were a number of barriers to activity, including a lack of agreement about the scope and remit of intervention, and whose priority it was to lead on these interventions. There was little specific national guidance to support decisions around how to manage inequalities in access and outcomes, but organisations could see the consequences of a lack of intervention on a daily basis. Although it was a legal duty, it was one of many competing priorities. Therefore, discussions about when, where and how to intervene often centred on questions around best use of available resource, and how impact of resource use could not be evidenced or tied to organisational key priorities.

Strengths of this study are that it is, to our knowledge, the first study to investigate staff's views on health inequalities in children's hospitals in England or elsewhere. The sample size of over 200 participants created a substantial dataset for qualitative analysis, which indicates that the findings are robust and transferrable.

Weaknesses include that participants were self-selecting and so may have volunteered to participate because they had a particular interest in sharing views on health inequalities. Rapid qualitative methods with selected transcription and analysis directly from audio recording were used, rather than full transcription of all interviews/focus groups.[21] This may be considered a weakness, but has been accounted for in the rigorous process conducted by the team, where recordings were reviewed multiple times by different researchers and a structured recording form was used to capture data of interest. As all data were recorded, there was opportunity to return to the transcripts as needed throughout the analysis process. Due to the remit of the funding for this research, it was not possible to involve patients or the public in the design, conduct, reporting or dissemination plans of our research, though our wider research has a

strong commitment to this aim. This meant we were also not able to speak to children, young people and families as part of this research, meaning their voices are represented by others within this dataset, which is a weakness of the study. Future research should address these questions around how health inequalities impact on healthcare from the perspective of children and young people in particular.

As identified in a recent scoping review protocol,[23] and the review itself,[24] there are few academic journal papers that outline activities and interventions conducted by children's hospitals to reduce health inequalities, and more transferrable evidence can be found in the grey literature (defined as sources of literature that have not been formally published). This is partly due to the more informal way that applied intervention and improvement work is often shared in the healthcare sector. Despite the acknowledged importance of reducing health inequalities for children and young people, little research has been conducted in the UK, and this is the first that has sought to understand how staff within NHS organisations perceive their role. This paper therefore makes an important contribution to understanding barriers and challenges around activities to reduce health inequalities and how hospital staff at all levels may contribute to these activities.

A recent review[25] aimed to outline 'the importance of healthcare workers advocating for structural and high-level policy change to address the deep-rooted societal problems that cause child poverty' by proposing a framework to address health inequalities experienced by children. Its broad conclusions support our more detailed account of the entrenched and intersectional nature of health inequalities and how they might be addressed within a children's hospital context. In a US context, similar questions are being raised about the potential role of children's hospitals in reducing health inequalities more usually addressed in a community setting.[26] A first next step may be to work directly with children and young people in future research to understand more about the challenges they face in relation to these health inequalities as experienced in everyday life.

## Conclusion and recommendations

While there is some clarity for organisations about their role in reducing health inequalities where they impacted on access and quality of care, there remains uncertainty about tackling health inequalities around the wider determinants of health, and the perceived boundaries of responsibility with other organisations within the wider health and social care system. In this study, most hospitals were forging ahead with activity on selected wider determinants, considering that it was more important to work to overcome health inequalities rather than debate whose job it was. While this is laudable, and the contribution of passionate individuals should not be underestimated, it does not position health inequalities clearly as part of children's hospitals' everyday work and so the sustainability

of interventions is not guaranteed. The needs of children and young people require much greater attention and stronger, clearer policy at a national level.

**Correction notice** This article has been corrected since it was published. Licence updated to CC BY on 2nd August 2024.

**Acknowledgements** Thank you to everyone at the hospital trusts who helped us arrange interviews and focus groups, and everyone who kindly gave up their time to contribute to our research. Special thanks to Anne Marie Davies, Children's Hospital Alliance and Alder Hey Children's Hospital, Rekel Kerr, Children's Hospital Alliance and Birmingham Children's Hospital and Alexandra Norrish, Children's Hospital Alliance and Sheffield Children's Hospital. Thanks to Tracy Briggs, Heather Catt, Fiona Egboko, Imelda Mayor, Nicola McCreddin, Pallavi Patel, Julie Pearcy, Dora Pestotnik Stres, Esther Primrose and Andrew Rowland for contributions to the wider project.

**Contributors** RI, LBrewster, LBrennan and JL conceptualised and designed the work. RI, LBrewster, LBrennan, AH, and JL collected the data with support from Broad, McCreddin, Patel and Pearcy (see acknowledgements). All authors analysed the data. LBrewster prepared the original draft. LBrennan, JL, AH, and RI critically reviewed and edited the work. All authors agreed the final version and are accountable for the accuracy and integrity of the work. RI is the guarantor for the study.

**Funding** This work was supported by the National Paediatric Accelerator Programme, now the Children's Hospital Alliance, contracted by Sheffield Children's Hospital, grant number SCH5628.

**Competing interests** The authors declare that they have no known competing financial interests or personal relationships that could have appeared to influence the work reported in this paper. The views expressed are those of the author(s) and not necessarily those of the Children's Hospital Alliance.

**Patient and public involvement** Patients and/or the public were involved in the design, or conduct, or reporting, or dissemination plans of this research. Refer to the Methods section for further details.

**Patient consent for publication** Not applicable.

**Ethics approval** Ethical approval was granted by FHM Research Ethics Committee Lancaster University on 16 June 2022 (ref: FHM-2022-0844-RECR-3). Health Research Authority approval was granted on 24 August 2022 (ref: IRAS315113 and 22/HRA/3123) and capacity and capability to participate were confirmed by each organisation.

**Provenance and peer review** Not commissioned; externally peer reviewed.

**Data availability statement** All data relevant to the study are included in the article or uploaded as supplementary information. Data from this study are not publicly available as per the terms and conditions of the funding of the study.

**ORCID iDs**
Liz Brewster http://orcid.org/0000-0003-3604-2897
Avni Hindocha http://orcid.org/0000-0003-2151-277X
Judith Lunn http://orcid.org/0000-0001-9281-2126
Rachel Isba http://orcid.org/0000-0002-2896-4309

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
