## [Reviewer comments · BMJ Open]

ARTICLE DETAILS

TITLE (PROVISIONAL)	Understanding responsibility for health inequalities in children's hospitals in England: a qualitative study with hospital staff
AUTHORS	Brewster, Liz; Brennan, Louise; Hindocha, Avni; Lunn, Judith; Isba, Rachel

VERSION 1 – REVIEW

REVIEWER	Buxton, Jane BC Centre for Disease Control, Harm Reduction
REVIEW RETURNED	29-Oct-2023

GENERAL COMMENTS	Thank you for the opportunity to review this interesting article: Understanding responsibility for health inequalities in children's hospitals in England: a qualitative study with hospital staff. The study methods are appropriate and engage with a variety of individuals. The findings could be presented in a more organized and structured way and discussion could focus on what the next steps should be. Abstract When did the interviews occur? Please include date in the abstract so it is clear when and over what period the interviews and focus groups occurred. In the methods it appears the data collection occurred in 2023 so including in the abstract would highlight how current the interviews were. It would be good to include a brief summary of the qualitative methods/approach used to analyse the results. The reader needs to be assured the opinions of 217 individuals are analysed in a systematic and rigorous way. Qualitative 'results' are usually called 'findings'. It would be easier to follow the 'results' if the authors used a framework or clearly organized way rather than free text to describe the findings. Introduction: The manuscript focusses on health inequalities in the title and introduction. In the first sentence of the introduction the authors describe health inequalities as unfair and avoidable. It would be good to have a broader explanation of why health inequalities rather than health inequities and the perceived differences. My understanding is that health inequities is a term used in reference to health inequalities that are unfair or unjust and are modifiable https://www.canada.ca/en/public-health/services/healthpromotion/population-health/what-determines-
---

	health.html The aim was described as to understand how children's hospitals view their responsibility to reduce health inequalities. I would suggest it is not the hospitals themselves that view the responsibility but the staff. Methods The qualitative approach was selected in order to describe participants' understanding. I would suggest qualitative approaches explores .. The hospitals were members of the Children's Hospital Alliance and recruitment occurred by the nine participating organisations. How many organizations are part of the Children's Hospital Alliance and how many Children's Hospital organizations are not part of the alliance, i.e. how representative were the nine participating organizations of all Children's Hospitals and the alliance? I note the protocol states ten children's hospitals but the study was performed n nine- what happened to #10 The authors refer to three broad categories of staff Leadership, Clinical and Professional and Support staff. However, it was not clear if representatives from all three categories participated in focus groups together or if the focus groups were composed of staff within similar categories. If the staff was mixed in the focus groups how did the researchers/facilitators deal with potential power dynamics? Table 1. The actual number of participants is usually included in results section rather than methods. The authors state variation of sites by various metrics but more detail regarding the organizations could be included either in table 1 or if concern re site identification in a descriptive text. Table 1 shows the number of interviews and focus group – although the reader can calculate the number of individuals in total who participated in the focus groups it would be clearer if the table included number of participants in focus groups at each site in the table. It would be helpful to know overall the number who participated in each of the three categories. Data collection Was there any incentive for participants e.g. an honorarium or opportunity to participate I a prize draw? Data analysis The analytic details are fairly explicit but it was not clear if the RAP sheet generation and discussions occurred after data collection was completed or if it was an iterative process and if the data review impacted any changes in the semi-structured interview or focus group guides. RESULTS The authors identified two main themes: Health inequalities as everyone's business and Health inequalities as 'no-one's responsibility. Under each theme there are a number of topics and quotes. Did
--	--

	these subthemes emerge purely from the data? Is there any pre-existing frameworks which correspond with your findings to compare /contrast with? A summary of what the header includes would be helpful to set the scene for what was coming next and organization into topic /subtheme areas would make it easier to read. Everyone's business Discuss strategies, responsibility and knowledge Challenges - lack of communication Supports – formal and informal Exacerbated by pandemic No one's responsibility Staff felt hospital has a role where inequalities impact access and quality of care Some felt could intervene, others not a priority and a few concerned re cost Challenges:  • lack of national and local priority; • unclear boundaries with other services • staff didn't feel part of the system • IT doesn't link health and social care systems • Champions may lead the work but often unnoticed and not sustainable if they left DISCUSSION The protocol mentions a scoping review, this is mentioned in the manuscript but findings of the review is not clear and seems little reference to what the finding of this study was and how reflects or differs from findings in the papers identified in the scoping review. In the discussion authors state the paper contributes to understanding activities to reduce health inequalities- I suggest it provides insights into barriers and challenges but I am less certain re activities to reduce health inequalities, There was some recognition of where individuals were taking initiatives but a lot of barriers to this being systemic. Minor comment: KPIs used only twice in the manuscript- suggest write in full both times
--	---

REVIEWER	Cronin, Cory Ohio University
REVIEW RETURNED	05-Nov-2023

GENERAL COMMENTS	Overall, this is a strong manuscript on an important topic. The perspective of the study adds to the value of the findings. I have a few points of relatively minor feedback that I feel would improve the paper for its readers. Primarily, it would be helpful to go into the policy description in greater detail, and then to reflect back on the details of the policy as they relate to the findings of the paper. Is there a policy path that the authors feel would make things clearer? Some additional details that would strengthen the paper:  - Clarification of how participants were consented in data collection - Citations supporting the "consensus that full transcription..." on p. 10 - Citation on p. 10, last paragraph, of work done in U.S. (Franz et al) on children's hospitals efforts to engage with social determinants
---

REVIEWER	Vadeboncoeur, Christina University of Ottawa
REVIEW RETURNED	07-Nov-2023

GENERAL COMMENTS	This manuscript addresses the understanding of leaders, clinical and support staff in Children's Hospitals in England about their responsibility to reduce health inequalities for the children and young people who access their services. It is an important contribution to the discussion of equity and equality in provision of health care, particularly for children.  1. The attached protocol outlines that 10 children's hospital will be contacted for the study, but the manuscript only speaks of 9 hospitals. It would be transparent to describe why there is a discrepancy in the number of participating institutions or why one institution was not included. 2. The authors state that the dataset can be considered a rich and robust representation of views within the nine organizations that they worked with, however their recruitment is described as having been organized by the nine participating organizations - and "guided by each one as to the most appropriate method to recruit participants". In addition, they describe three broad categories of staff (leadership, clinical and professional and support staff, but then state they conducted semi-structured interviews with senior leaders (leadership) and doctors with an interest in health inequalities (clinical) and then state they conducted focus groups and had informal conversations with clinical, professional and support staff. It is not clear that this provides a robust representation of views. 3. Although briefly mentioned in the discussion, the lack of involvement of children and families in this report is a major limitation and means their voices are lacking, rather than that their voices are represented by others as stated by the authors.
---

VERSION 1 – AUTHOR RESPONSE

Reviewer: 1

Dr. Jane Buxton, BC Centre for Disease Control, University of British Columbia

Comments to the Author:

Thank you for the opportunity to review this interesting article: Understanding responsibility for health inequalities in children’s hospitals in England: a qualitative study with hospital staff. The study methods are appropriate and engage with a variety of individuals. The findings could be presented in a more organized and structured way and discussion could focus on what the next steps should be.

Thank you for your comments. We have taken feedback on the results and discussion on board and tried to incorporate these comments without significantly increasing the word count.

Abstract

When did the interviews occur? Please include date in the abstract so it is clear when and over what period the interviews and focus groups occurred. In the methods it appears the data collection occurred in 2023 so including in the abstract would highlight how current the interviews were.

This has been added.

It would be good to include a brief summary of the qualitative methods/approach used to analyse the results. The reader needs to be assured the opinions of 217 individuals are analysed in a systematic and rigorous way.

A sentence has been added to the abstract: Analysis: RREAL methodology for rapid assessment procedures. There is not scope to include more in the abstract. Further information on analysis has been added in the main data collection section.

Qualitative 'results' are usually called 'findings'.

We were guided by the submission guidelines and termed our findings accordingly. We are happy to take a steer from the editor on how to refer to these findings.

It would be easier to follow the 'results' if the authors used a framework or clearly organized way rather than free text to describe the findings.

The results section of the abstract has been restructured to clarify the organisation of the results. Given the nature of the findings it was not appropriate to use a framework for analysis. We have added further structure to the start of each thematic section.

Introduction:

The manuscript focusses on health inequalities in the title and introduction. In the first sentence of the introduction the authors describe health inequalities as unfair and avoidable. It would be good to have a broader explanation of why health inequalities rather than health inequities and the perceived differences. My understanding is that health inequities is a term used in reference to health inequalities that are unfair or unjust and are modifiable <https://www.canada.ca/en/public-health/services/health-promotion/population-health/what-determines-health.html>

In the UK, there is a tendency to use inequality as a 'catch all' term to describe unjust differences, while recognising this is not the preference on the north American continent (e.g. -

The aim was described as to understand how children's hospitals view their responsibility to reduce health inequalities. I would suggest it is not the hospitals themselves that view the responsibility but the staff.

We have clarified as suggested.

Methods

The qualitative approach was selected in order to describe participants' understanding. I would suggest qualitative approaches explores ..

This has been changed as suggested.

The hospitals were members of the Children's Hospital Alliance and recruitment occurred by the nine participating organisations. How many organizations are part of the Children's Hospital Alliance and how many Children's Hospital organizations are not part of the alliance, i.e. how representative were the nine participating organizations of all Children's Hospitals and the alliance?

A sentence has been added to state that:

At time of the study, the Children's Hospital Alliance had ten members; it now has eleven. As a national informal network of specialist NHS trusts focused on paediatric care in the UK, it initially focused on support for pandemic recovery for children and young people.

While we can state that the nine organisations were representative of the Alliance, it is very hard to clarify how representative the membership is of children's hospitals in the UK, as criteria for membership does not rest on being a 'standalone' children's hospital. This diversity has also been noted in the earlier discussion (in the introduction) of how children's hospitals in the UK are organised.

I note the protocol states ten children's hospitals but the study was performed in nine- what happened to #10

A sentence has been added to state that: One of the ten was unable to participate due to capacity.

The authors refer to three broad categories of staff Leadership, Clinical and Professional and Support staff. However, it was not clear if representatives from all three categories participated in focus groups together or if the focus groups were composed of staff within similar categories. If the staff was mixed in the focus groups how did the researchers/facilitators deal with potential power dynamics?

A sentence has been added to state that:

Focus groups were composed of staff from similar categories to ensure that those present felt more comfortable to contribute.

Table 1.

The actual number of participants is usually included in results section rather than methods.

This was also discussed between the co-authors. However, the lead author takes the view that for qualitative work 'number of people who participated' is not a finding – it is part of the data collection process and so sits in methods not results.

We have left this as written for now and are happy to take a steer from the editor on this.

The authors state variation of sites by various metrics but more detail regarding the organizations could be included either in table 1 or if concern re site identification in a descriptive text.

The included hospitals have concerns re site identification and so we have agreed not to describe them further. As there are now eleven members of this organisation and we worked with nine of them, we appreciate their concerns re identifiability. The full list of member organisations is easily found via internet searching, and so saying (e.g.) 'leadership, large hospital in north of England' would make it possible to identify potential participants.

If this is not acceptable to the editor, we are prepared to revisit this with the sites. We would appreciate a steer from the editor on this.

Table 1 shows the number of interviews and focus group – although the reader can calculate the number of individuals in total who participated in the focus groups it would be clearer if the table included number of participants in focus groups at each site in the table.

This has been clarified as suggested.

It would be helpful to know overall the number who participated in each of the three categories.

This has been clarified as suggested.

Data collection

Was there any incentive for participants e.g. an honorarium or opportunity to participate in a prize draw?

There were no incentives to participate.

Data analysis

The analytic details are fairly explicit but it was not clear if the RAP sheet generation and discussions occurred after data collection was completed or if it was an iterative process and if the data review impacted any changes in the semi-structured interview or focus group guides.

A sentence has been added to state that:

RAP sheet generation occurred simultaneously with data collection, with discussion occurring after data collection had been completed.

RESULTS

The authors identified two main themes: **Health inequalities as everyone's business** and **Health inequalities as 'no-one's responsibility**. Under each theme there are a number of topics and quotes. Did these subthemes emerge purely from the data? Is there any pre-existing frameworks which correspond with your findings to compare /contrast with?

A summary of what the header includes would be helpful to set the scene for what was coming next and organization into topic /subtheme areas would make it easier to read.

Everyone's business

Discuss strategies, responsibility and knowledge

Challenges - lack of communication

Supports – formal and informal

Exacerbated by pandemic

No one's responsibility

Staff felt hospital has a role where inequalities impact access and quality of care

Some felt could intervene, others not a priority and a few concerned re cost

Challenges:

- lack of national and local priority;
- unclear boundaries with other services
- staff didn't feel part of the system
- IT doesn't link health and social care systems
- Champions may lead the work but often unnoticed and not sustainable if they left

These themes and subthemes were identified purely from the data, and we are not aware of any pre-existing frameworks that could be used to compare the findings with. We appreciate the reviewer's comment regarding additional summary structure and have added in as suggested.

DISCUSSION

The protocol mentions a scoping review, this is mentioned in the manuscript but findings of the review is not clear and seems little reference to what the finding of this study was and how reflects or differs from findings in the papers identified in the scoping review.

The protocol is for the full study as commissioned, rather than being a specific protocol for this arm of the study.

At time of writing, the scoping review protocol had been completed, but the scoping review results were not available. The scoping review has now been completed and is under review with BMJ Open. To avoid duplication of reporting of results, have added a reference to this review rather than outlining its findings. The scoping review focused on what activities children's hospitals conducted, rather than what staff thought about these activities, so there is little crossover in relevance. For reference, the scoping review questions were: What approaches do hospitals take to address health inequalities in children and young people? What health inequalities do the approaches focus on? How is effectiveness measured and demonstrated?

In the discussion authors state the paper contributes to understanding activities to reduce health inequalities- I suggest it provides insights into barriers and challenges but I am less certain re activities to reduce health inequalities, There was some recognition of where individuals were taking initiatives but a lot of barriers to this being systemic.

This has been edited as suggested.

Minor comment:

KPIs used only twice in the manuscript- suggest write in full both times

This has been edited as suggested.

Reviewer: 2

Dr. Cory Cronin, Ohio University

Comments to the Author:

Overall, this is a strong manuscript on an important topic. The perspective of the study adds to the value of the findings. I have a few points of relatively minor feedback that I feel would improve the paper for its readers. Primarily, it would be helpful to go into the policy description in greater detail, and then to reflect back on the details of the policy as they relate to the findings of the paper. Is there a policy path that the authors feel would make things clearer?

Thank you for this helpful comment. One of our key drivers for writing this article based on our analysis of the data from staff was that the national policy outlines an idea of duties but lacks detail. This was in contrast with other measurable and measured targets and competing priorities (referenced as key performance indicators in the UK, and often associated with extra payments etc), which meant that there was no clear steer on what attention should be paid to the duty to 'reduce inequalities in access and outcomes' or financial incentive to do so.

We have added a sentence to the introduction to make this clearer:

These top-level strategies focus on a legal duty to reduce inequalities in access, and outcomes for all without specific guidance on how this may be implemented.

We have also reflected back on this in the discussion.

Some additional details that would strengthen the paper:

- Clarification of how participants were consented in data collection

This has been edited as suggested.

- Citations supporting the "consensus that full transcription..." on p. 10

We have expanded our section on how this was done, with reference to rapid appraisal methods more broadly, as we felt this was more reflective of our position and approach to the data, and more useful to the reader.

- Citation on p. 10, last paragraph, of work done in U.S. (Franz et al) on children's hospitals efforts to engage with social determinants

This has been added as suggested.

Reviewer: 3

Dr. Christina Vadeboncoeur, University of Ottawa

Comments to the Author:

This manuscript addresses the understanding of leaders, clinical and support staff in Children's Hospitals in England about their responsibility to reduce health inequalities for the children and young people who access their services. It is an important contribution to the discussion of equity and equality in provision of health care, particularly for children.

1. The attached protocol outlines that 10 children's hospital will be contacted for the study, but the manuscript only speaks of 9 hospitals. It would be transparent to describe why there is a discrepancy in the number of participating institutions or why one institution was not included.

As above, this has been clarified.

2. The authors state that the dataset can be considered a rich and robust representation of views within the nine organizations that they worked with, however their recruitment is described as having been organized by the nine participating organizations - and "guided by each one as to the most appropriate method to recruit participants".

We have clarified that while organisations called for volunteers using their preferred route (e.g. mailing list post, social media) and helped to identify appropriate staff to approach by role (e.g. the

chief nurse, medical director etc of each organisation) they did not select participants per se. This has been revised in the text accordingly.

In addition, they describe three broad categories of staff (leadership, clinical and professional and support staff, but then state they conducted semi-structured interviews with senior leaders (leadership) and doctors with an interest in health inequalities (clinical) and then state they conducted focus groups and had informal conversations with clinical, professional and support staff. It is not clear that this provides a robust representation of views.

For qualitative research, having a sample of participants from nine different organisations comprising input from over 200 different individuals can be considered a robust dataset. However, we accept that this is different from a robust representation of views and have removed this sentence.

3. Although briefly mentioned in the discussion, the lack of involvement of children and families in this report is a major limitation and means their voices are lacking, rather than that their voices are represented by others as stated by the authors.

We agree that the lack of inclusion of children and young people is a weakness. We have clarified our reference to this in the discussion to highlight this. In this instance, the research was delivered in response to a commissioned call which focused on understanding staff views and so there was no option to include them here. We have referenced our wider ongoing programme of research which aims to be more inclusive.

VERSION 2 – REVIEW

REVIEWER	Buxton, Jane BC Centre for Disease Control, Harm Reduction
REVIEW RETURNED	13-Dec-2023

GENERAL COMMENTS	Thank you for addressing the reviewers comments and providing clear responses to each suggestion METHODS  - Thank you for providing more detail re composition of participants at sites and roles. - For completeness I suggest Table 1, 3rd column includes # participants in header and in total line # participants in FG/conversations. - Patient and public involvement- says occurred concurrently with this study but may be misleading as I don't believe this input is incorporated in this manuscript. It is addressed in discussion weaknesses but would suggest some explanation up front so reader is not misled into thinking patient and public involvement occurred in this study. RESULTS "... two main themes were identified that help to understand how children's hospitals.. view their responsibility". As in abstract objectives and first sentence in discussion I believe it is staff in the hospitals who view their responsibilities Thank you for qualifying the theme headers – sets up the description in a better way DESCRIPTION A sentence has been added "This is partly due to the way that applied intervention and improvement work is often shared in the healthcare sector." But it's not clear what 'the way' means.
---

VERSION 2 – AUTHOR RESPONSE

Thank you for these comments and suggestions. We have:

- Revised Table 1, 3rd column as suggested.
- Clarified the input from our PPI work
- Altered the results/ abstract as suggested to clarify that we are referring to staff.
- Clarified the sentence re improvement work.